# Development of Computational In Silico Model for Nano Lipid Carrier Formulation of Curcumin

**DOI:** 10.3390/molecules28041833

**Published:** 2023-02-15

**Authors:** Omar Waleed Abduljaleel Albasri, Palanirajan Vijayaraj Kumar, Mogana Sundari Rajagopal

**Affiliations:** Faculty of Pharmaceutical Sciences, Department of Pharmaceutical Technology, UCSI University, Jalan Menara Gading, Taman Connaught, Cheras, Kuala Lumpur 56000, Malaysia

**Keywords:** SLNs, NLCs, curcumin, drug carrier, permeability, molecular docking

## Abstract

The oral delivery system is very important and plays a significant role in increasing the solubility of drugs, which eventually will increase their absorption by the digestive system and enhance the drug bioactivity. This study was conducted to synthesize a novel curcumin nano lipid carrier (NLC) and use it as a drug carrier with the help of computational molecular docking to investigate its solubility in different solid and liquid lipids to choose the optimum lipids candidate for the NLCs formulation and avoid the ordinary methods that consume more time, materials, cost, and efforts during laboratory experiments. The antiviral activity of the formed curcumin–NLC against SARS-CoV-2 (COVID-19) was assessed through a molecular docking study of curcumin’s affinity towards the host cell receptors. The novel curcumin drug carrier was synthesized as NLC using a hot and high-pressure homogenization method. Twenty different compositions of the drug carrier (curcumin nano lipid) were synthesized and characterized using different physicochemical techniques such as UV–Vis, FTIR, DSC, XRD, particle size, the zeta potential, and AFM. The in vitro and ex vivo studies were also conducted to test the solubility and the permeability of the 20 curcumin–NLC formulations. The NLC as a drug carrier shows an enormous enhancement in the solubility and permeability of the drug.

## 1. Introduction

Since the early 1990s, pharmaceutical technology research groups have been paying increasing attention to lipid nanoparticles for diverse applications as carrier systems [1,2]. The researchers looked at solid lipid nanoparticles (SLNs) and nanostructured lipid carriers (NLCs) [3]. Different NLCs formulations, including curcumin, were developed for oral use in this work. NLCs are new colloidal carrier systems composed of solid and liquid lipids. These lipids are carefully combined to create particle–matrix mixtures [4]. Compared to pure solid lipids, the presence of liquid lipids in these mixtures produces depression with a melting point. The resulting blends are solid at room and body temperature [5,6]. NLCs are a new kind of lipid nanoparticles that have the advantages of improving the drug loading and storage capacities within the particle matrix [7]. The medicine during storage, presented as the loading capacity, was enhanced. Changing the lipid concentration of NLCs by the incorporation of liquid lipid resulted in improving the physical stability. However, drugs that have a higher solubility in oils than in solid lipids can be dissolved in the oil, and the role of the surrounding lipid is to provide the chemical stability for active compounds that are chemically sensitive [8].

A variety of lipids, both solid and liquid, are used in the manufacturing process of SLNs and NLCs formulations [9]. Palmitic acid, sitosterol alcohol, glyceryl monostearate (GMS), tripalmitate, and stearic acid are the solid lipids which are often used in SLNs and NLCs formulations [10]. Liquid lipids such as oleic acid, olive oil (containing ethyl palmitate), corn oil (containing apocarotenal), and grape seed oil (containing linoleic acid) are combined with solid lipids to make NLCs. Normally, tween 80 and poloxamer 188 are typically used as surfactants [11].

Curcumin is mostly derived from the natural plant. The richest and most cost-effective source of crude curcumin is *Curcuma longa* [12]. It has also been used to treat a range of problems in various patients, including diabetes, liver disease, cancer, and rheumatoid disease [13]. Curcumin is only sparsely soluble in water [14]. Chemically, it is an unstable molecule with a limited biological half-life because it is poorly absorbed and rapidly metabolized [15]. All these factors contribute to curcumin’s lower bioavailability when taken as a whole [16]. It is categorized as GRAS (generally recognized as safe) by the US Food and Drug Administration [17]. The most attractive and vital reason for the therapeutic use of curcumin is its superior safety profile. It has been demonstrated that curcumin has a very low toxicity [18]. The bioactivity of orally taken curcumin has been the most fundamental basis for this apprehension. Thus, it can be used as an anti-inflammatory, anti-microbial, anti-oxidant, anti-viral, and anti-cancer agent [19,20,21].

In the last two decades, much research and many experiments were carried out to investigate the optimal characteristics of the drug [22]. The previous trials were not efficient and expensive [23], thus virtual screening was applied as a new approach based on structural information [24]. The method of virtual screening can be classified as structure-based and ligand-based drug designing methods [25,26]. The first one describes molecular docking while the second method deals with the relationship between the quantitative structure’s activity and pharmacophore modeling [27,28]. The structural information increases the range of molecular targets of proteins and protein–ligand complexes via certain steps, starting with chemical synthesis techniques, purification, X-ray crystallography, and nuclear magnetic resonance spectroscopy (NMR) [29].

Generally, the molecular docking method determines the interaction between the ligand and target molecule [30,31]. It can also predict the ligand binding affinity in the formation of a stable complex with protein in minimum free binding energy [32,33]. The binding can occur via non-covalent interactions such as hydrogen bonds, ionic bonds, hydrophobic, and van der Waals forces [34].

The current research proposed the development of NLC molecules to improve the oral solubility and permeability of curcumin. The research also focused on developing a computational molecular docking to study the solubility of curcumin in different solid and liquid lipids to choose the optimum candidates for the NLCs formulation rather than the frequent experimental method, which consumes much time, efforts, and high costs. The molecular docking model was applied to study and enhance the antiviral activity of curcumin against COVID-19 by studying the binding affinity to the host cell receptors of the angiotensin-converting enzyme (ACE2) [35].

## 2. Results and Discussion

### 2.1. Molecular Dynamic Study for Curcumin Solubility and Interaction with Candidate Lipids

The docking study of curcumin with lipids was performed through PyRx software. Eleven types of lipids: solid lipids (GMS, stearic acid, palmitic acid, sitosterol alcohol, and tripalmitate) and liquid lipids (oleic acid, apocarotenal, ricinoleic acid, ethyl palmitate, linoleic acid, and omega-3) were utilized to perform the docking study for the solubility and interaction prediction between curcumin and the selected lipids. These were previously tested and evaluated through laboratory experiments (solubility and partition study of curcumin with lipids). Then, a comparison of the obtained results from the docking study with the one obtained from the laboratory experiment to specify the degree of accuracy, validity, and reliability for the applied docking system for solubility anticipation between the curcumin and lipids was made.

For apocarotenal (obtained from corn oil) CID 5478003, the types of bond interaction between apocarotenal and curcumin were examined. The results represent one hydrogen bond as a carbon–hydrogen bond (distance 3.507) and six hydrophobic pi–alkyl bonds (distance range 3.849–4.897). For omega-3 (obtained from soybean oil) CID 71306824, the types of bond interaction between omega-3 and curcumin were examined. The results represent one hydrogen bond as a conventional hydrogen bond (distance 2.398), one hydrogen bond as a carbon–hydrogen bond (distance 3.537), and three hydrophobic pi–alkyl bonds (distance range 5.066–5.472). For linoleic acid (obtained from grape seeds oil) CID 5280450, the types of bond interaction between linoleic acid and curcumin were examined. The results represent one hydrophobic pi–sigma (distance 3.573) bond and three hydrophobic pi–alkyl bonds (distance range 3.792–4.982). For ethyl palmitate (obtained from olive oil) CID 12366, the types of bond interaction between ethyl palmitate and curcumin were examined. The results represent one hydrophobic pi–sigma (distance 3.536) bond and four hydrophobic pi–alkyl bonds (distance range 3.658–5.498). For ricinoleic acid (obtained from castor oil) CID 14030006, the types of bond interaction between ricinoleic acid and curcumin were examined. The results represent three hydrophobic pi–alkyl bonds (distance range 3.879–5.082) as shown in Figure 1.

Oleic acid shows the types of bond interaction between oleic acid and curcumin. The results represent four hydrogen bonds (distance range 3.464–3.514) and twelve hydrophobic pi–alkyl bonds (distance range 3.911–5.433). Figure 2 represents the 3D interaction between the curcumin molecule and the oleic acid molecule with bond types and distances.

Considering solid lipids (SL), the types of bond interaction between Cetostearyl alcohol and curcumin were examined. The results represent four hydrophobic pi–alkyl bonds (distance range 4.015–4.306), while the types of bond interaction between stearic acid and curcumin were examined. The results represent four hydrophobic pi–alkyl bonds (distance range 3.827–5.029). The types of bond interaction between palmitic acid and curcumin were examined, and the results represent two hydrogen bonds (distance 3.545 and 3.643) and four hydrophobic pi–alkyl bonds (distance range 3.654–5.101). The types of bond interaction between tripalmitin and curcumin were examined and the results represent two hydrogen bonds as conventional hydrogen bonds (distance 2.664 and 2.701) and two hydrophobic pi–alkyl bonds (distance 3.917 and 4.729), as presented in Figure 3.

As observed in Table 1, the types of bond interaction between GMS and curcumin represent six hydrogen bonds as conventional hydrogen bonds, one hydrophobic pi–sigma (C–H) bond, and one hydrophobic pi–alkyl bond (pi-orbital). Figure 4 represents the 3D and 2D interaction between the curcumin molecule and GMS molecule, respectively; the bonds type and distances were also described.

The scale of root means square deviation (RMSD) clustering for the solubility construal docking for GMS was set to nine and was used to determine how well scoring combinations pose and calculate the ligand (CUR) in the lipid site (GMS). Discovery Studio software was used to interpret the obtained docking results. Molecular docking insights revealed that CUR bound to GMS via six hydrogen bonds, one pi–sigma C–H hydrophobic bond, and one pi–alkyl hydrophobic bond. The values for the binding energy of each lipid with curcumin were obtained and used as an indicative tool to expect the stability of the formed formulations and was approximately −8.6 Kcal/mol for GMS, which was significantly lower than the other compared ten lipids (range from −6.5 to −1.8).

The lowest binding energy of GMS has the highest binding affinity to curcumin and indicates that the ligand binds strongly to the receptor, which proves that GMS has the highest solubility tendency for curcumin. BIOVIA Discovery Studio software confirmed the interaction between CUR and GMS. It was reported that conventional hydrogen bonding shows the strongest interaction between molecules and the results showed that six conventional H-bonding were generated between CUR as the H-donner and GMS as the H-accepter in the type of OH–H bond, which shows fundamental solubilizing services in a biomolecular structure [36].

Hydrogen bonds perform as facilitators to CUR–GMS binding and offer an inordinate chance for the solubilization of CUR in GMS since it can endorse the ligand binding affinity among the donor and acceptor. It was conveyed that hydroxyl groups of GMS play a crucial role in binding through hydrogen bonding generation and solubilizing CUR. Therefore, this demonstrated that CUR could bind and have an affinity to GMS through strong conventional hydrogen bonding [37]. In addition, the shortest distance of these conventional hydrogen bonds compared to another type of bond present (less than 3000 °A) proves that the presence of such bonds gives a higher binding affinity and solubility between curcumin and GMS molecules [38]. These findings were not reported with the other compared 10 lipids. Moreover, the presence of a pi–alkyl bond in the interaction between curcumin and GMS provides the stability for the system, and the presence of the pi–sigma bond provides an extra stability to the interaction between the molecules [39]. On the other hand, oleic acid was found to be the best liquid lipid candidate to be mixed with GMS for the NLCs formulation since the high compatibility between these two lipids was found experimentally and due to the obtained docking results, which showed that oleic acid was superior compared to other liquid lipids in demonstrating a lower binding energy of −6.5 kcal/mol. Moreover, the presence of four C–H bonds in the oleic acid–curcumin interaction allows solubility [40].

A comparison between the predicted solubility of curcumin in different solid and liquid lipids obtained from the molecular docking study (through calculating the binding energy) and the actual solubility results obtained from laboratory experiments (by calculating the amount of curcumin dissolved in these lipids) was conducted to see how far this computational docking system can be recommended for choosing the optimum solid and liquid lipid. The results were promising and showed a similar order of solubility (from highest to lowest) for both solid and liquid lipids, indicating the accuracy of the docking study applied, as shown in Table 2.

### 2.2. Formulation of Curcumin-Loaded Nanostructured Lipid Carriers

The influence of the process parameters, homogenization pressure, and speed is very important for the production of NLC by the hot high-pressure homogenization (HHPH) method [41]. The correct choice of both the pressure and speed would directly affect the size of the nanoparticles [42]. For this purpose, the pressure range was taken in between 500 and 1000 bars, and the speed range was 10,000–15,000 rpm. To optimize the formulation of curcumin NLC, the dependent variables including the entrapment efficacy (EE%) particle size, ploy dispersity index (PDI), and zeta potential were evaluated to choose the best formula. Table 3 shows the results of the formula obtained concerning the EE% P size, PDI, and zeta potential. The outcomes showed that the particle size of the best selected formulas (F1,F2,F4,F12, and F19) was less than 200 nm, which lies within the acceptable range [43], the PDI was found to be ≤0.3, indicating a unimodal or a uniform mono dispersion size distribution [44], the zeta potential results were more than 40, indicating a good stability [45], and the EE% results were high, meaning that the majority of the drug added during the formulation process was entrapped, resulting in less loss of valuable active medicinal ingredients [46].

### 2.3. Compatibility Study of the Excipients Used in the NLCs Formulations

#### 2.3.1. Fourier Transforms Infrared Spectra

Curcumin’s characteristic FTIR peaks were identified at 3502 cm^−1^, indicating the presence of the phenolic OH group. The prominent peak at 1629 cm^−1^ was identified as a mixed peak of (C=C) and (C=O) character vibrations. The symmetric aromatic ring stretching vibrations (C=C ring) have another high peak at 1602 cm^−1^. The strong 1512 cm^−1^ peak is attributed to (C=O) olefinic C–H bending vibrations at 1344 cm^−1^, while an enol C-O band was produced at 1280 cm^−1^, a C–O–C peak at 1114 cm^−1^, and benzoate trans-C–H vibration at 962 cm^−1^. Figure 5 represents the FTIR spectrum of GMS and oleic acid. In the FTIR spectra of the prepared curcumin NLCs-F1, there was no discernible change in curcumin’s distinctive peaks (Figure 6). Furthermore, there was no evidence of peak shifting.

The lack of additional peaks/broadening of the peaks suggests that the process parameters did not cause the curcumin to be stressed, resulting in degradation. Curcumin also had no chemical interactions and was confirmed to be compatible with the formulation excipients [47,48].

#### 2.3.2. Differential Scanning Calorimetry

At 180.55 °C, pure curcumin’s melting endothermic peak was observed that complied with similar findings [49,50]. Moreover, other formula excipients like lipids, ethanol, poloxamer 188, and tween 80 showed a thermogram that proves the compatibility of curcumin with all excipients used for the formulation. By detecting the fluctuation of the temperature and energy during the phase transition, DSC was an approach for investigating the crystallization or amorphous behavior of medication in pure and NLCs. The melting process for PRE reached a maximum peak temperature of 60.93 °C. The melting peak for the CUR, on the other hand, was not visible in the thermogram of the lyophilized CUR–NLCs with the presence of an oleic acid peak at 240 °C. The CUR entrapment in hydrophobic voids of GMS could explain the lack of a distinctive endothermic peak in the DSC thermogram of NLCs. Figure 7 showed the DSC thermogram of the selected formula F1 in comparison to pure curcumin. The removal of CURs endothermic peak in CUR–NLCs powder shows that the drug was molecularly disseminated in the lipid matrix and transformed from a crystalline to an amorphous state [47,51].

#### 2.3.3. X-ray Diffraction Spectra (XRD)

The XRD patterns of pure drug and freeze-dried drug-loaded NLCs F1 were investigated. Curcumin peaks were absent or had a very low or negligible intensity in NLCs, indicating a decline in freeze-dried curcumin’s crystallinity. NLCs with broad and low-intensity peaks have a weak crystalline character. This was due to curcumin’s trapping within the NLCS. The XRD analysis of the curcumin shows that the pure substance has a very high crystallinity with an intense sharp peak. The crystallinity of the drug was significantly reduced in NLCs, indicating curcumin entrapment within the lipidic core. This may have been achievable because of curcumin’s high solubility in the solid lipid (GMS) which was utilized [52]. Figure 8 shows the XRD analysis of the selected formula F1 in comparison to pure curcumin.

### 2.4. Evaluation of Curcumin–NLC Formulation

#### 2.4.1. Determination of Drug Content and Incorporation of Efficacy

The NLCs were broken using methanol, and the encapsulated drug was recovered for examination. The curcumin incorporation efficacy of the formulations investigated was moderate to high, ranging from 45.32 ± 0.28% to 83.45 ± 0.13% (*w*/*w*) with the highest content found in the optimum F1 84%, which was considered an acceptable percent [53,54]. The % medication concentration in all the NLCs formulations ranged from 85.67 ± 2.34 to 95.68 ± 2.38.

#### 2.4.2. Particle Size and Poly-Dispersity Index (PDI)

The PDI of curcumin-loaded optimized NLCs revealed a system that was displaying substantial polydispersity. It displays the PDI of 0.192 and the particle size of the F1 formulation (99.64 nm). Because of their non-ionic nature, polyhydroxy surfactants stabilize the formulation by generating a spatial exclusion, resulting in low and nearly PDI with the highest particle dispersion [55]. A PDI score of 0.10 to 0.40 indicates that the system is considerably poly-dispersed [56]. Moreover, the low particle size provides a high surface area of contact with the dissolution medium, resulting in the enhanced solubility of curcumin [57].

#### 2.4.3. Zeta Potential

The results of the zeta potential for curcumin-loaded optimized NLCs F1 was −42.30 ± 0.01, indicating a good stability behavior of the colloids. Surfactant, lipid, or combined surfactant and lipid may be responsible for the surface charge. Tween 80 and poloxamer 188 were utilized as the surfactants and stabilizer in the manufacture of NLCs. Tween 80 is a nonionic surfactant, which means it does not affect the particle surface charge (zeta potential). Poloxamer 188, on the other hand, is a nonionic block linear copolymer-type stabilizer, and its negative charge contributes to the particle polarity. The lipid phase of the NLCs was made up of GMS and oleic acid. GMS is a long-chain fatty acid glycerol ester that does not affect the surface charge, whereas oleic acid is a medium-chain fatty acid whose negative surface charge related to the carboxyl group may contribute to the zeta potential [58]. As a result, a zeta potential of more than −40 mV reflects a good stability behavior of the prepared colloid system [59].

#### 2.4.4. EE Percent and Drug Loading

When the amount of GMS was raised in curcumin-NLCs, the EE improved. However, it was dropped when the oleic acid concentration was reduced, and the EE was shown to be lower. This may be related to lipid precipitation, which happens during the creation of particles. When NLCs are cooled after being made, the lipids recrystallized, resulting in a drug-free core or a core with a reduced drug content. As a result, a lipid rise beyond a certain point causes a poor EE. Tween 80 was also found to have a substantial effect. The EE of the formulation rose as the concentration of tween 80 was increased. The EE of the NLCs dispersion was found to be F1, F2, F4,and F12, and F19 had a percentage EE of 84.23 ± 1.35, 80.11 ± 0.60, 83.41 ± 0.29, 83.08 ± 0.13, and 82.31 ± 0.27%, respectively. A factorial design was applied to optimize the prepared NLCs and assess their critical quality attributes through EE [60]. Figure 9 shows that the increase in the lipid concentration and media time sonication will increase the EE. The response surface central composite was developed (CCD) in design expert software (Version 10.0.1, Stat-Ease Inc., Minneapolis, MN, USA) was used to design the experiment, which fitted to the quadradic model using Factorial Design-Expert software [61].

### 2.5. In Vitro Drug Release

The curcumin release potential from the lipid particles was tested for 24 h. A spectrophotometric approach was used to analyze each sample in triplicate. In vitro tests were conducted to compare the release properties of pure curcumin and the NLCs (Figure 10). The time taken for an aliquot time interval of medication to be released showed that the dissolution rate was increased in NLCs. Formula F1 had a burst release pattern at first, with more than 50% of the medication being released within 6 h, whereas pure curcumin took more than 24 h to dissolve 50% of the curcumin. When compared to other formulations, NLCs formulation F1 had the smallest particle size when compared to pure curcumin and the other formulations F2, F4, F12, and F19, which showed a higher release rate in the dissolution media. As a result, this formulation was chosen for further testing [62,63].

The in vitro drug release research was conducted at 37 °C for 24 h in a phosphate buffer (pH 6.8). Because curcumin is a lipophilic medication, it was added to the receptor medium to keep the sink conditions constant. The data clearly show that NLCs released the medication significantly more efficiently than a simple curcumin suspension. This can be ascribed to the phospholipidic nano formulations’ nanosized particles. Nanoparticles have a larger surface area, which improves the particle interaction with the dissolving liquid. Due to the release of curcumin adsorbed on the surface of nanoparticles, all of the curcumin-loaded NLCs showed a burst drug release over the first 4 h. Following that, the release of curcumin from the nanoparticles showed a regulated release pattern, with roughly 85% of the medication being released in the F1 formulation for up to 12 h, compared to ordinary curcumin. It could be attributed to the fact that curcumin, which is contained deep within the lipidic core and must travel along a longer diffusion path to reach the surface than a medication encapsulated near the surface, is hydrophobic [64]. The increased drug release from the F1 formulation compared to the other formulations (F2, F4, F12, and F19) could be attributed to the much smaller particle size, which increases the surface area and, as a result, the drug release rate [65,66].

### 2.6. Morphological Analysis

The AFM image was obtained with a scan rate of 0.5 Hz over a selected area in the dimension of 3 × 3 µm. The force applied to the surface was roughly adjusted by a set-point of 16 nm and an amplitude of 26.57 nm. An atomic force microscopy examination in 3D imaging presented a round spherical shape of the prepared nanoparticles [67]. The particle size in the AFM micrographs of nanoparticles is somewhat bigger than in the dynamic light scattering micrographs, which is likely owing to the flattening of the NLCs during the drying phases in the sample preparation for AFM imaging (Figure 11) [68]. The phase image confirmed the stability of the NLCs formulation due to the bright surfaces, which indicates positive phase shift–repulsive forces between the tip and samples, suggesting the good hydration of the lipid [15,69]. Curcumin-loaded nanoparticles had a similar size of 99.64 ± 8.64 nm. The attendance of a spheroidal shape, closer to a disc than a sphere, is confirmed by the reduced nanoparticle height reported by AFM. As previously stated, this structure validates the frequency of the polymorphic form of lipids, which is linked to a high loading capacity and a low tendency to expulse the encapsulated drug from the lipid matrix [70].

### 2.7. In Vitro Gut Permeation Study

Figure 12 depicts the intestinal permeability of a curcumin suspension and curcumin-NLCs. The permeability of a curcumin suspension and curcumin NLCs through the rat gut was reported to be 6.83 ± 1.53 (Papp of 0.488 × 10^−5^ cm/s) and 42.52 ± 3.15 mcg/cm^2^/h (Papp of 6.84 × 10^−5^ cm/s), respectively, after 8 h. At *p* ≥ 0.05, there was a substantial difference in the permeability between the curcumin NLCs formulation and curcumin suspension. Smaller particle sizes and the incorporation of permeation enhancers (poloxamer 188 and tween 80) resulted in an increased drug penetration via curcumin NLCs, according to these findings [71].

### 2.8. Molecular Docking Studies of the Anti-Viral Activity of Curcumin NLC against SARS-CoV-2 (COVID-19)

The docking scores of the binding affinity between the different host cell receptors and curcumin in the curcumin–NLC complex are tabulated in Table 4. A comparison was applied for the obtained docking results of the curcumin interaction with the host cell receptors of ACE 2 and the curcumin interaction with NLC in the curcumin–NLC–host cell receptor complex to study the influence of the formed curcumin–NLC on the binding affinity for the target host cell receptors. First, the PyRx software was applied for the docking of curcumin with the NLC components and it was then explored through the Discovery Studio visualizer. Again, the obtained micelles were subjected to docking with the three host cells through the same process as described earlier.

The anti-viral potential of curcumin against COVID-19 using different targets was investigated. The results showed a significant reduction in the binding energy, indicating an enhancement in the binding affinity of the ligand toward the targets. The curcumin interaction with NLC in a complex has shown an interaction potential and exhibited −6.8 kcal/mol binding energy and formed one conventional H-bond with C: ASP994, one carbon–hydrogen bond C: GLY999, two pi–sigma C: THR998 and A: THR998, and five pi–alkyl bonds B: VAL991, C: PHE970, A: TYR756, B: ARG995, and C: ARG995. Table 5 and Table 6 epitomize the results of the data obtained from the active residues, bond length (A0), bond types, and bond categories involved in the molecular interactions of curcumin with NLC and curcumin with different receptors (7KMB,7KNB, and 7KNH) in the curcumin–NLC–receptor complex.

For the 7KMB receptor, the curcumin interaction with the receptor in a complex represents a significant enhancement in the binding affinity through an energy reduction to −9.1 kcal/mol; the interaction also formed an additional four H-bonds compared to the two H-bonds formed for curcumin with NLC, indicating an increase in the curcumin affinity towards the receptor. The formed conventional H-bond showed a great increase through six formed bonds F: TYR202, F: SER511, F: SER511, F: ARG514, F: LYS562, and F: GLY395. Moreover, pi–anion, pi–pi stacked, and pi–alkyl were formed with F: ASP206, F: TYR202, and F: TYR202, respectively (Figure 13).

For the 7KNB receptor, interestingly, the curcumin interaction with the receptor in a complex represents a strong enhancement of the binding affinity through an energy reduction to −8.6 kcal/mol and formed a new extra five H-bonds, indicating a high increase in the curcumin affinity towards the receptor. The interaction showed six conventional H-bonds C: THR998, A: ARG995, A: ARG995, A: ARG995, B: ARG995, and B: VAL991, one carbon–hydrogen bond C: ASP994, one amide pi-stacked bond A: ASP994:C, O;ARG995, and 3 pi–alkyl bonds C: VAL991, A: ARG995 and B: ARG995 (Figure 14).

For the 7KNH receptor, a study of the curcumin interaction with the receptor in a complex represents an enhancement in the binding affinity through an energy reduction to −8.4 kcal/mol and formed new additional six H-bonds and carbon–hydrogen bonds, indicating a high increase in the curcumin affinity towards the receptor. The interaction showed five conventional H-bonds C: GLN755, B: ASN969, B: ARG995, B: ARG995, and B: GLY971, three carbon–hydrogen bonds C: ASP994, B: ALA972, and B: ILE973, one pi–anion bond C: GLU990, and one pi–alkyl bond C: LEU752 (Figure 15).

The results in Table 6 show a significant tendency of the binding affinity of curcumin to the three binding receptors of ACE2 when formulated as an NLC due to the low binding energy and the increased H-bonding [72]. Consequently, these increase the antiviral property of curcumin against SARS-CoV-2 and disrupt SARS-CoV-2 binding to ACE2 binding receptors and prevent their entry into the cell. It is anticipated that enhancing the curcumin solubility through the NLCs formulation will provide a suitable concentration of the medicine for its absorption and bioavailability, which enhances the curcumin’s anti-viral activity in the treatment of COVID-19 [73].

## 3. Materials and Methods

### 3.1. Materials

Curcumin, stearic acid, palmitic acid, and sitosterol alcohol were obtained from Sigma-Aldrich, St. Louis, MO, USA. Glyceryl monostearate (GMS), poloxamer 188, and tween 80 were obtained from Research-lab Fine Chem Industries, Mumbai, India. Oleic acid and tripalmitin were kindly received as gifts from TCS Chemicals, Mumbai, India.

### 3.2. Molecular Dynamic Study for Curcumin Solubility and Interaction with Solid and Liquid Lipids

The interaction of curcumin (CUR) lipids for the solubility prediction [SDF, PDB] was studied. Low molecular weight curcumin was derived from the PubChem website (https://pubchem.ncbi.nlm.nih.gov/, accessed on 3 January 2023) and downloaded as SDF and converted to pdb format using Discovery Studio Software (version- 19.1.0.18287). Eleven lipid structures including solid lipids (GMS, tripalmitin, stearic acid, palmitic acid, and sitosterol alcohol) and liquid lipids (oleic acid, apocarotenal, ethyl palmitate, omega-3, ricinoleic acid, and linoleic acid) were also downloaded as SDF from the PubChem website and prepared for docking through the same process as curcumin. All the structures of the selected compounds that aided as molecules for modeling studies were adjusted using the Auto-Dock software (PyRx software version 0.8, python.exe) before docking. Using PyRx software, the lipid molecules were subjected to auto-dock to build a macromolecule.

The ligand curcumin was dragged to the software in SDF format and subjected to minimizing energy (E = 234.8) and was saved as the pdbq format. After selecting each lipid molecule involved (which was used in experimental laboratory studies in this project) and ligand, run vina was applied to study the interaction between lipid and ligand curcumin through PyRx software. The docking outcomes were scrutinized to recognize and assess the binding capability between these compounds. Discovery Studio software was used to analyze the type of bond interaction between lipid and curcumin and we chose the best fit interaction for the optimum solubility. Auto-dock vina in PyRx software is the most preferable software for molecular docking. The molecular docking approach can be used to model the interaction between small molecules and proteins at the atomic level, allowing for the characterization of the behavior of small molecules in the binding site of target proteins, as well as elucidating the fundamental biochemical processes [74,75]. Discovery Studio software is an agglomeration to transcribe small molecules and macromolecule systems. It is developed by Dassault Systems BIOVIA (Accelrys) [76]. Discovery Studio is a single unified, graphical interface for advanced drug design and protein modeling research. This software provides a plethora of viewers for display plots and graphical representations of data [77]. In the present study, there was an investment in this software to predict the interaction and solubility between hydrophobic drugs such as curcumin and lipids.

### 3.3. Formulation of Curcumin-Loaded Nanostructured Lipid Carriers

A hot high-pressure homogenization method was used to make NLCs. The first step was to mix 200 mg of glycerol monostearate with 200 mg of oleic acid at 80 °C. Then, 10 mg of curcumin was mixed at the same temperature, dispersing a solution of poloxamer 188 (1% *w*/*v*) and tween 80 (1% *w*/*v*). The surfactant phase was achieved by heating the lipids utilized in the study over their melting point. For 15 min, a machine that stirred at 15,000 rpm mixed the lipids and surfactants. It was homogenized three times at 1000 bar pressure and 80 °C. A nanoemulsion was made and when it cooled, it turned into a mixture of lipid nanoparticles in water. Different formulas were prepared for the NLCs production using high-pressure homogenization and ultra-sonication to prepare 20 formulas, as shown in Table 7. A total of 10 mg of curcumin was used in all the formulas with different lipids percent (GMS and oleic acid) using different amounts of surfactants (poloxamer 188 and tween 80) along with a different stirring time.

### 3.4. Compatibility Study of the Excipients Used in the NLCs Formulations

For the estimation of the compatibility among the component of the NLCs formulation, FT-IR, DSC, and XRD studies were carried out. In this study, the spectra of all the individual ingredients and the NLCs formulation were taken. The IR spectra were taken by an FT-IR instrument (OPUS, Bruker, ALPHA, Billerica, MA, USA) in a region of 400 to 4000 cm^−1^ and with a number of scanning of 32. A few amounts of the powder samples (further grinded for homogenization) were placed in thin KBr disks and compressed using a minipresss machine to prepare them for reading. Differential scanning calorimeter (ARKS TA, PerkinElmer, DSC 4000, Waltham, MA, USA) curves were collected. For the calibration of the instrument, indium was used. Approximately 10 mg of the powder sample was taken in an alumina crucible and placed in the DSC. Then, over a temperature range of 20 °C to 520 °C within a nitrogen rich environment, it was heated at a rate of 10 °C/min. The DSC thermograms were then visually analyzed [78]. An X-ray diffractometer was used to analyze the samples during a period of 2 h with an angle of 10° to 80°. The X-ray diffraction (XRD) examination of the NLCs was carried out using SPEC, Bruker model D8 ADVANCE X-ray diffractometer (Hamburg, Germany), using Cu-K radiation as the source of X-rays at a 40 kV working voltage and a 250 W electricity set was employed. At a 5°/min scanning speed and 0.02° step size, the samples were scanned in the range of 3–50° ± 2°.

### 3.5. Evaluation of Curcumin NLC Formulation

#### 3.5.1. Drug Content

When 10 mg equivalent NLCs were dispersed for 15 min in triton X-100 (0.1% in methanol) and centrifuged at 15,000 rpm for 30 min, the amount of medicine in the dispersion was assessed. Additionally, the visible spectrum at 425 nm was found after being diluted with methanol. A Shimadzu-LC 2030 HPLC and a standard graph of the drug produced in methanol were used to measure the drug concentration. The HPLC system is equipped through a Shimadzu LC-10AD VP pump, a C18 Agilent column (Inertsil, 150 mm × 4.66 mm, 5–VIS detector), and a flow rate of 1 mL/min; the mobile phase has been degassed. It is made up of an acetonitrile, water, and methanol (50:10:40) ratio. The detector wavelength was set at 425 nm. The injection volume was 10 µL. Methanol was used as the diluent.The lipid-specific NLCs blanks were all the same [79].

#### 3.5.2. Particle Size and Particle Size Distribution (PDI)

The polydispersity index (PDI) and mean particle size were measured using the dynamic light scattering of NLCs F1-NLCs F20 (DLS Zetasizer, Nano ZS Malvern, Worcestershire, UK). Non-invasive backscatter technology was used to detect light scattering at 173° (detector angle), measuring the particle size in a range of approximately 0.6 nm to 6 µm. A foldable polycarbonate capillary cell was used to conduct the measurement in a 50 V applied field. Before the analysis, to achieve a homogeneous dispersion, the dispersions were diluted (1:200) in 0.22 m filtered double-distilled water. Three different batches of curcumin NLCs were used to determine the mean particle size and PDI for each concentration. The mean particle size standard deviation of three determinations was obtained for each sample. The entire analysis was conducted at a temperature of 30 °C.

#### 3.5.3. Zeta Potential (ζ)

The zeta potential, also known as the surface charge potential, was estimated using the Horiba nanoparticle size analyzer (SZ-100 nanoparticle series). Diluted NLC dispersions were injected into the probe using an 80 mV electric field electrophoretic cell. All measurements were made in triplicate at 25 °C. The zeta potential was then directly determined from the equation by using the Smolochowski equation as follows [80,81].
ζ = εµ/η
where ζ is the zeta potential, µ is the electrophoretic mobility, ε is the electric permittivity of the liquid, and η is the viscosity of the liquid.

#### 3.5.4. Entrapment Efficiency and Drug Loading

The Entrapment efficiency EE can be calculated using either the encapsulated or the unloaded bioactive. This can be done with a hydrophilic bioactive by centrifugation for 10 min at 13,000 rpm, which yields an unloaded bioactive filtrate, which can be used to determine the EE.


$$\%\;Entrapment\;efficiency=\frac{\left(weight\;of\;drug\;added-free\;drug\right)}{\left(weight\;of\;drug\;added\right)}\;\times \;100$$


### 3.6. In Vitro Drug Release Study

In vitro drug release studies on pure curcumin and curcumin NLCs were carried out to determine the best formulation for further study. The dialysis bag’s molecular weight threshold of 12–14 KDa was used to determine the in vitro drug release from the formulations [35,82]. In this experiment, 10 mg of pure curcumin and the selected best five NLCs formulas were inserted into a dialysis bag, which was suspended in a 250 mL beaker filled with 100 mL of synthetic intestinal fluid (pH 6.8, phosphate buffer, tween 80) and placed on a magnetic stirrer that spun at 100 revolutions per minute and was maintained at a temperature of 37 °C. Regularly, new dissolution medium samples were taken from the dissolution medium and replaced. A volumetric flask filled with 10 Ml of methanol was treated with 1 mL of the filtrate after it was filtered through Whatman filter paper No. 1. As needed, additional dilutions were made. A spectrophotometric technique was used to analyze the materials. Calculations were made with the Korsmeyer and Peppas model and the PCP Disso Version 2.08 programmed [83,84,85,86,87,88] to determine the time needed for 50 and 85% drug release, respectively.

### 3.7. NLCs Morphology Determination by AFM

Atomic force microscopy (AFM) was used to obtain high-resolution 3D images and it provides more information about a sample’s mechanical and electrical properties, such as the stiffness and adhesion strength. The samples were not coated. AFM was used to study the morphology and particle size of the formed NLCs [89]. The studies were conducted in contact mode in the air at an ambient temperature (25 °C). On a tiny mica disc, droplets of the final suspension (20 µL) were deposited and rinsed with Milli-Q water and dried under a nitrogen flow after 5 min. The measurements were carried out in non-contact mode using the high-resonance AFM cantilever (ACTA probe, n = 330 kHz) on an XE-100 instrument [90].

### 3.8. In Vitro Gut Permeation Study

The experimental protocol was approved by the Institutional Animal Ethical Committee (IAEC) with approval number CPCSEA/IAEC/JLS/16/9/21/13. Experiments on a fasted Wistar rat’s (weighing 250–300 g) small intestine were conducted in a laboratory setting. An intestinal segment was removed from a dead rat after euthanasia using CO_2_ inhalation and it was cut into lengths of 6–7 cm. Curcumin-NLCs and curcumin suspension were added to the Krebs Ringer solution used to flush the stomach, and the other end of the syringe was used to inject 1 mL of each formulation into the Sac. In a beaker heated to 37 °C, the sac was submerged in an oxygenated Krebs Ringer solution (20 mL). The sink was maintained by removing and replacing aliquots of 1 mL every 0.5, 1, 1.5, 2, and 8 h with an equivalent volume of warm Krebs Ringer solution heated to 37 °C. Each aliquot’s medication concentration was measured using HPLC (n = 3). The HPLC system is equipped through a Shimadzu LC-10AD VP pump, a C18 Agilent column (Inertsil, 150 mm × 4.66 mm, 5–VIS detector), and a flow rate of 1 mL/min; the mobile phase has been degassed. It is made up of an acetonitrile, water, and methanol (50:10:40) ratio. The detector wavelength was set at 425 nm. The injection volume was 10 µL. The samples were measured after diluting suitably with methanol. Each formulation’s flow and permeability coefficients were calculated [91]. The Curcumin permeability coefficient (Papp) was computed from the mucosal to the serosal direction using the equation:


$$Papp\left(\frac{cm}{sec}\right)=\left(\frac{DQ}{dt}\right)\left(AxCO\right)$$


The *DQ/dt* is the rate of drug permeation from the tissue, *A* is the cross-sectional part of the tissue, and *Co* is the initial curcumin concentration in the donor compartment at *t* = 0.

### 3.9. Molecular Docking Studies of the Anti-Viral Activity of Curcumin against SARS-CoV-2 (COVID-19)

The molecular docking was conducted on the Lenovo ThinkPad T440p using the PyRx-Virtual Screening Tool. The structure of curcumin (sdf file format) was downloaded from the official website of the National Center for Biotechnology Information PubChem (https://pubchem.ncbi.nlm.nih.gov/, accessed on 3 January 2023). The energy minimization (optimization) was performed by a Universal Force Field (UFF). The 3D structure of curcumin–NLC constituents GMS, oleic acid, poloxamer 188, and tween 80 were obtained from drawing these chemical structures through ChemSchetch software and transferred from 2D to 3D and saved (as.mol file format). Then, they were converted to PDB format using the Discovery Studio visualizer 2019.

These structures were prepared for docking through PyRx software. The three host cell receptor structures of ACE2 (PDB ID: 7KMB, 7KNB, and 7KNH) were obtained from the RCSB PDB site (https://www.rcsb.org/, accessed on 3 January 2023). The receptor structures, with the aid of the Discovery Studio Visualizer 2021, were optimized, purified, and prepared for molecular docking. Autodock vina 1.1.2 in PyRx 0.8 was used to perform the molecular docking studies. A large number of glycosylated S proteins cover the surface of SARS-CoV-2 and bind to the host cell receptor angiotensin-converting enzyme 2 (ACE2), mediating a viral cell entry. Once the virus enters the cell, the viral RNA is released. Polyproteins are translated from the RNA genome, and the replication and transcription of the viral RNA genome occurs via the protein cleavage and assembly of the replicase–transcriptase complex. Viral RNA is replicated, and structural proteins are synthesized, assembled, and packaged in the host cell, after which viral particles are released [92]. For molecular docking, both ligands (PDBQT files), as well as the targets, were selected. The active binding receptors were examined and lightened using the BIOVIA Discovery Studio Visualizer (version-19.1.0.18287).

### 3.10. Statistical Analysis

The results are expressed as the mean ± standard deviation (SD) (n = 6), and the statistical analysis was performed with GraphPad Prism 8 (GraphPad Software, Inc., La Jolla, CA, USA). The significance difference between the experimental groups was evaluated by the one-way ANOVA. The counter and response surface plots of the entrapment efficiency (sonication time and lipid concentration) was evaluated using a quadric model (factorial design).

## 4. Conclusions

The computational molecular docking denotes that the binding energy for GMS was very low (−8.6 Kcal/mol), which indicates the highest binding affinity to curcumin and a strong binding to the receptor with the highest solubility tendency for curcumin. Therefore, glyceryl monostearate (GMS) was chosen as the best solid lipid with the highest drug solubility among the screened solid lipids. Oleic acid was selected as a liquid lipid for the NLCs formulation due to its high binding affinity to curcumin and compatibility with GMS as well. The major absorption peaks of the FTIR with a prominent peak were observed at 1629 cm^−1^ as identified by the prepared NLC-curcumin. The reduced melting point in the DSC detection and the XRD results shows a lower peak intensity of the curcumin–NLC formula, indicating the amorphization of the curcumin crystals.

The low particle size of the novel NLC-curcumin provides a high surface area of contact with the dissolution medium, which enhanced the solubility of curcumin. The optimum NLCs formulas had an entrapment efficacy percent (EE%) = 84.23 ± 1.35. The AFM image confirmed the spherical shape and the low particle size of the prepared NLC. The novel NLC-curcumin shows an increase in the drug release from the best formula compared to the other formulations. This could be attributed to the much smaller particle size, which increases the surface area and, as a result, the drug release rate was increased with the suspension of curcumin-NLCs. The permeability and penetration release rate in the intestine and the gut was reported after 8 h and it was found to be 6.83 ± 1.53 and 42.52 ± 3.15 mg/cm^2^/h, respectively, due to the small size of NPs and to the presence of permeation enhancers (poloxamer 188 and tween 80). The future perspectives of the study include the requirements of determination of the industrial feasibility of the proposed delivery system, scale-up, and pilot plant studies.

## Figures and Tables

**Figure 1 molecules-28-01833-f001:**
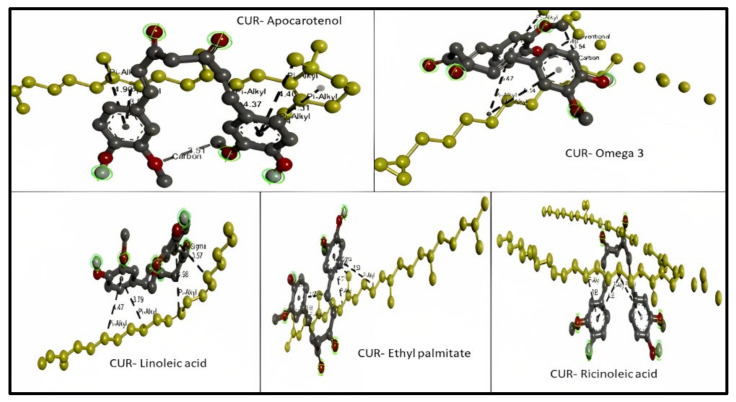
3D molecular interaction between curcumin (CUR) with different liquid lipids (highlighted in yellow) for solubility prediction. The formed bonds are described as black short lines.

**Figure 2 molecules-28-01833-f002:**
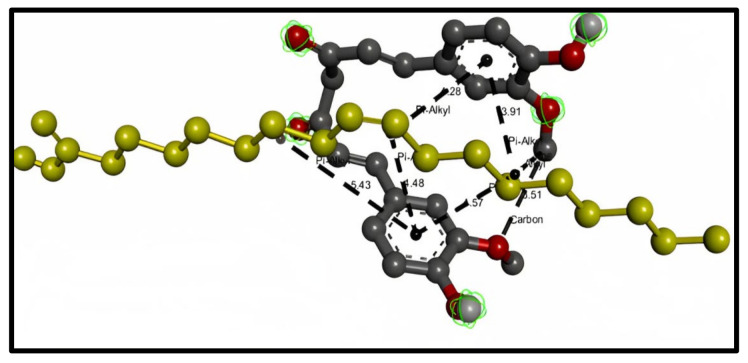
3D interaction between curcumin molecule (gray) and oleic acid oil (yellow). The formed bonds are described as black short lines.

**Figure 3 molecules-28-01833-f003:**
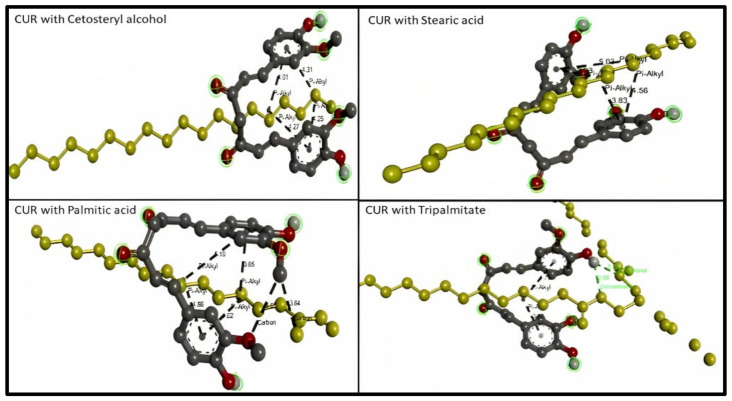
3D molecular interaction between curcumin (CUR) with different solid lipids (highlighted in yellow) for solubility prediction. The formed bonds are described as black short lines.

**Figure 4 molecules-28-01833-f004:**
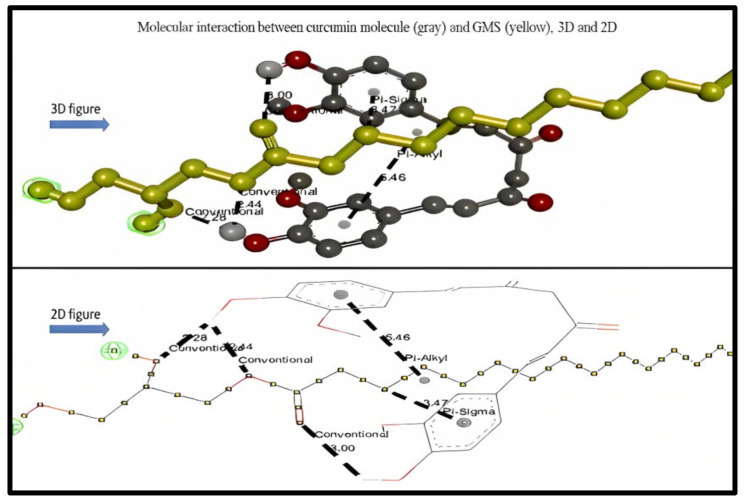
3D and 2D molecular interaction between curcumin molecule (gray) and GMS (yellow).

**Figure 5 molecules-28-01833-f005:**
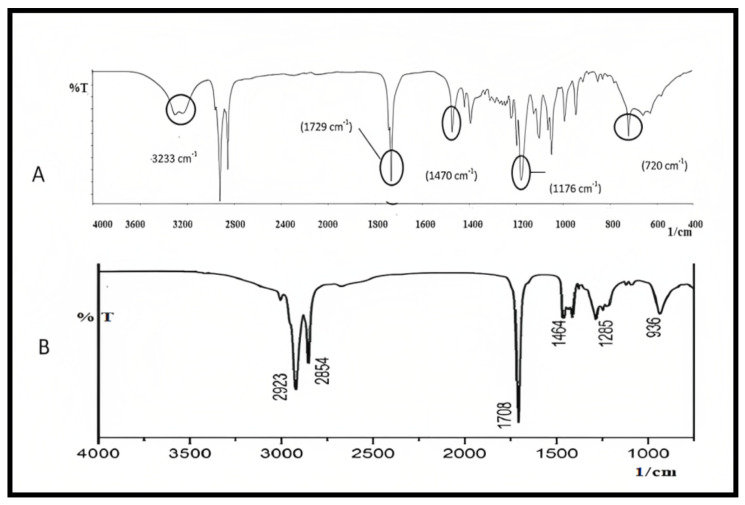
FTIR spectra of (**A**) GMS and (**B**) oleic acid, plot of %Transmittance vs. 1/cm.

**Figure 6 molecules-28-01833-f006:**
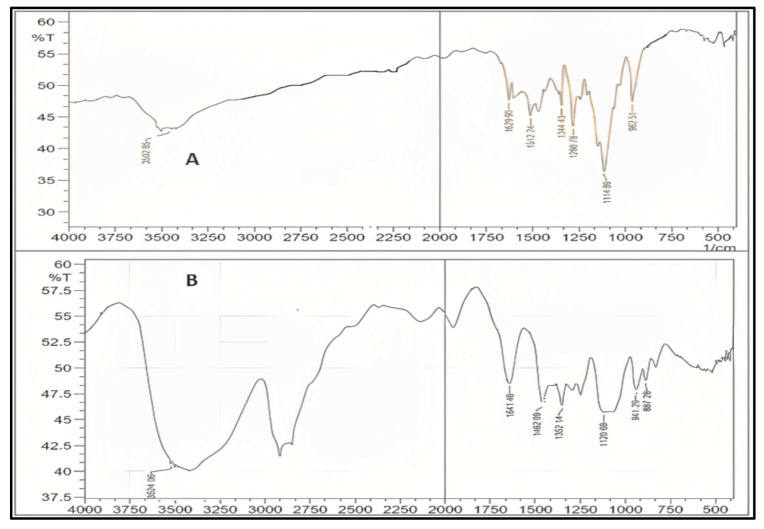
FTIR for pure curcumin (**A**), the optimized formulation of curcumin–NLC F1 (**B**), which shows no chemical interaction with curcumin major peaks, plot %Transmittance vs. 1/cm.

**Figure 7 molecules-28-01833-f007:**
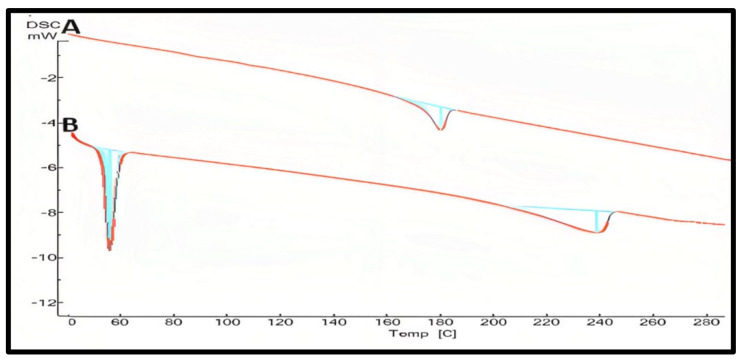
DSC studies of pure curcumin (**A**), the prepared curcumin NLCs-F1 (**B**) with the absence melting peak of curcumin at 180 °C confirming amorphous state plot of mW vs. temp.

**Figure 8 molecules-28-01833-f008:**
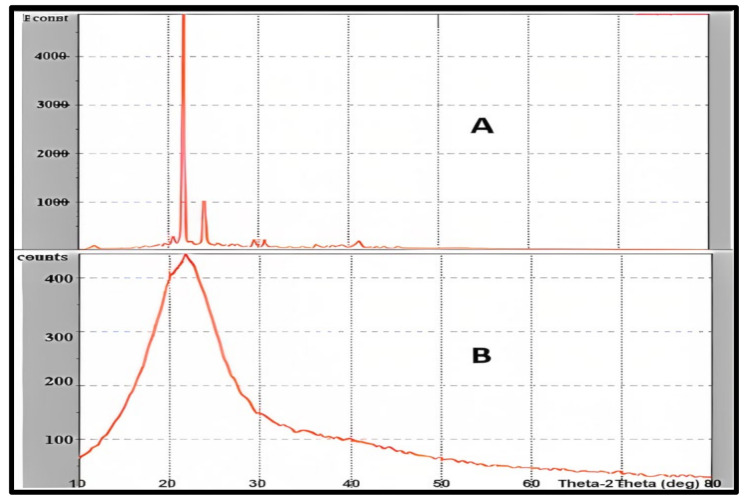
Powder X-ray diffraction studies of pure curcumin (**A**), and curcumin NLCs-F1 (**B**) presenting significant lowering intensity due to high amorphization, the plot of Counts vs. Theta-2 Theta (deg).

**Figure 9 molecules-28-01833-f009:**
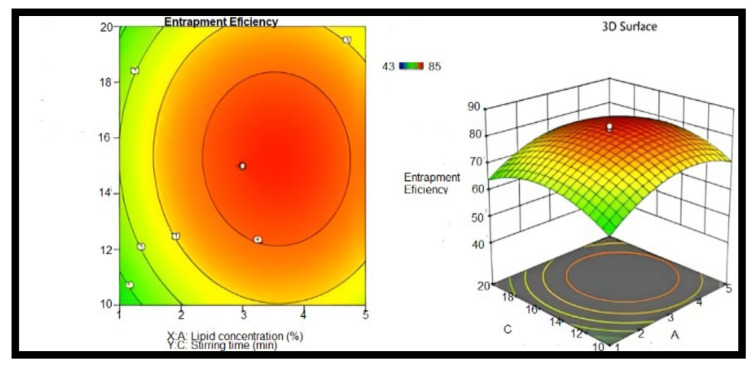
Counter and response surface plots of entrapment efficiency (stirring time and lipid concentration), n = 6.

**Figure 10 molecules-28-01833-f010:**
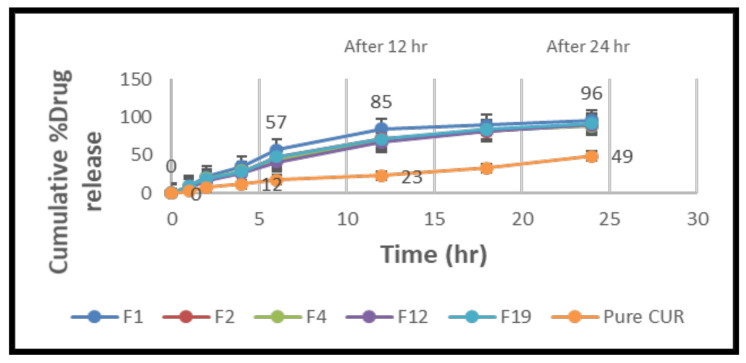
Comparison of cumulative % drug release NLC formulas (F1, F2, F4, F12, and F19) and pure curcumin suspension versus time (h), mean ± SD (n = 6).

**Figure 11 molecules-28-01833-f011:**
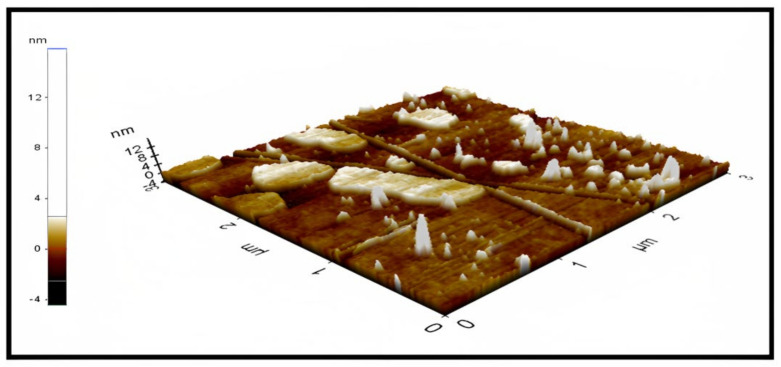
Atomic force microscopy image of optimum curcumin–NLC F1 determining the morphology of the prepared nano lipids (size area 3 × 3 µm).

**Figure 12 molecules-28-01833-f012:**
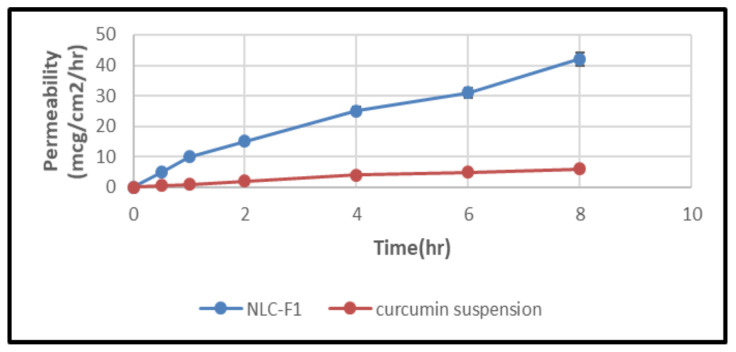
Intestinal permeation study representing permeation of curcumin suspension and curcumin-NLCs versus time (h), mean ±SD (n = 6).

**Figure 13 molecules-28-01833-f013:**
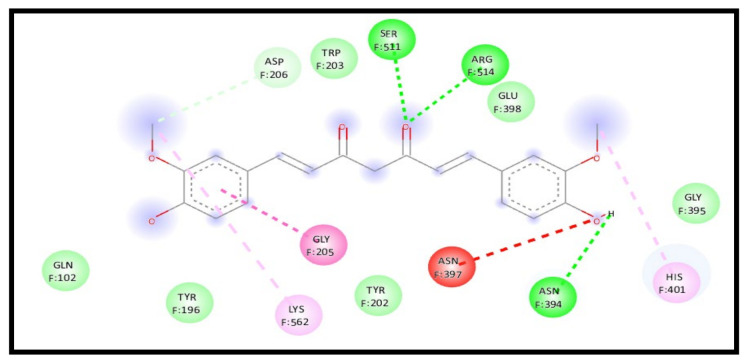
2D interaction of curcumin–NLC with ACE 2 binding receptor (7KMB) in a complex.

**Figure 14 molecules-28-01833-f014:**
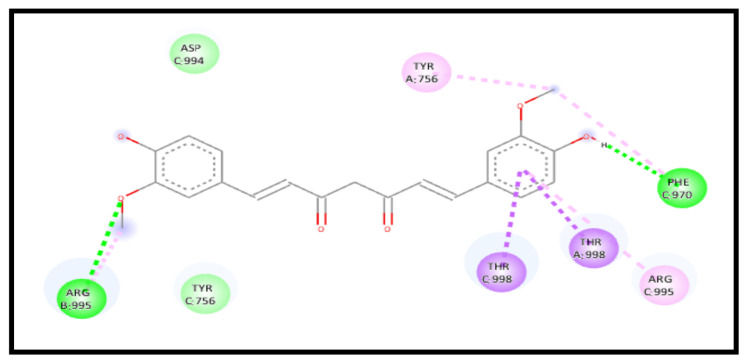
2D interaction of curcumin–NLC with ACE 2 binding receptor (7KNB) in a complex.

**Figure 15 molecules-28-01833-f015:**
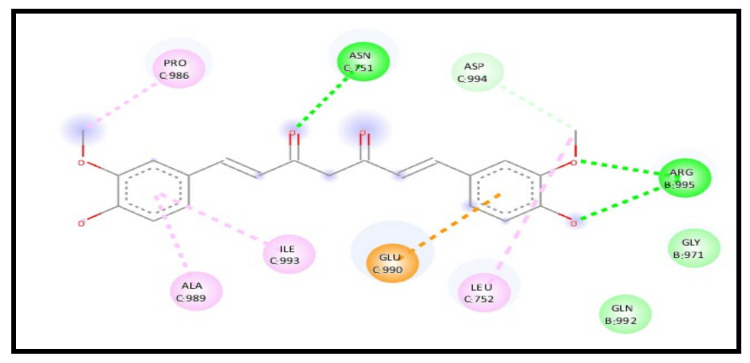
2D interaction of curcumin–NLC with ACE 2 binding receptor (7KNH) in a complex.

**Table 1 molecules-28-01833-t001:** Types of bonds for curcumin–GMS interaction.

Distance	Category	Type
2.44011	Hydrogen bond	Conventional hydrogen bond
2.27678	Hydrogen bond	Conventional hydrogen bond
2.99625	Hydrogen bond	Conventional hydrogen bond
3.46712	Hydrophobic	Pi–sigma
5.46068	Hydrophobic	Pi–alkyl
2.44011	Hydrogen bond	Conventional hydrogen bond
2.27678	Hydrogen bond	Conventional hydrogen bond
2.99625	Hydrogen bond	Conventional hydrogen bond

**Table 2 molecules-28-01833-t002:** Predicted versus actual results of curcumin solubility in different lipids.

Solid Lipids	Predicted Solubility (kcal/mol)	Actual Solubility (mg/g)
GMS	−8.6	3.58 ± 0.03
Palmitic acid	−4.8	3.28 ± 0.06
Tri Palmitin	−2.9	3.25 ± 0.30
Stearic acid	−2.4	2.62 ± 0.12
Cetosteryl	−1.8	2.06 ± 0.13
**Liquid lipids**		
Oleic acid	−6.5	4.61 ± 0.13
Castor oil	−5.3	3.52 ± 0.52
Olive oil	−5.1	3.48 ± 0.34
Soybean oil	−4.7	1.36 ± 0.26
Corn oil	−4.4	1.06 ± 0.15
Grape seeds oil	−3.9	0.59 ± 0.64

**Table 3 molecules-28-01833-t003:** Formulations results of EE%, particle size, PDI, and zeta potential.

F. Code	EE%	P. Size (nm)	PDI	Zeta (-)
**F1**	84.23 ± 1.35	99.64 ± 8.64	0.192 ± 0.01	42.3 ± 0.01
**F2**	80.11 ± 0.60	100.29 ± 5.16	0.172 ± 0.03	41.8 ± 0.13
**F3**	56.37 ± 2.13	164.06 ± 3.49	0.694 ± 0.06	23.4 ± 0.11
**F4**	83.41 ± 0.29	99.86 ± 4.16	0.164 ± 0.29	41.8 ± 0.34
**F5**	61.62 ± 3.17	190.64 ± 2.31	0.524 ± 0.17	29.1 ± 0.06
**F6**	70.09 ± 0.84	159.10 ± 3.19	1.135 ± 0.34	26.4 ± 0.08
**F7**	55.25 ± 3.73	305.04 ± 1.05	0.961 ± 0.19	24.7 ± 0.46
**F8**	49.19 ± 2.49	341.23 ± 2.43	0.659 ± 0.03	22.9 ± 0.61
**F9**	45.64 ± 0.51	251.11 ± 1.38	0.654 ± 0.24	20.4 ± 0.94
**F10**	60.73 ± 0.28	284.32 ± 0.67	0.829 ± 0.06	30.4 ± 0.31
**F11**	65.34 ± 0.61	268.06 ± 4.13	0.675 ± 0.14	34.9 ± 0.62
**F12**	83.08 ± 0.13	123.29 ± 5.03	0.197 ± 0.28	41.8 ± 0.09
**F13**	68.09 ± 1.28	146.16 ± 1.29	0.829 ± 0.07	29.1 ± 0.37
**F14**	52.43 ± 1.42	193.43 ± 3.45	0.753 ± 0.03	32.5 ± 0.13
**F15**	62.39 ± 3.24	227.16 ± 4.07	0.761 ± 0.31	29.7 ± 0.16
**F16**	81.06 ± 1.06	116.22 ± 3.18	0.219 ± 0.62	41.8 ± 0.08
**F17**	49.43 ± 2.67	237.24 ± 5.10	0.894 ± 0.37	28.1 ± 0.27
**F18**	52.57 ± 2.61	249.70 ± 0.94	0.691 ± 0.09	20.6 ± 0.31
**F19**	82.31 ± 0.27	110.13 ± 0.07	0.168 ± 0.15	39.7 ± 0.82
**F20**	43.51 ± 3.16	312.13 ± 1.49	0.694 ± 0.20	25.1 ± 0.19

**Table 4 molecules-28-01833-t004:** The docking scores of curcumin with different targets (host cell receptors) in the curcumin–NLC complex.

Target	Binding Energy (kcal/mol) for ACE2 Receptors–Curcumin in Complex
ACE 2	
**7KMB**	−9.1
**7KNB**	−8.6
**7KNH**	−8.4

**Table 5 molecules-28-01833-t005:** The active residues, bond length (A0), bond types, and bond categories involved in the molecular interactions of curcumin with NLC in a complex.

Bond Length (A0)	Bond Category	Bond Type
2.90749	Hydrogen bond	Conventional hydrogen bond
3.5423	Hydrogen bond	Carbon hydrogen bond
3.2796	Hydrophobic	Pi–Sigma
3.8657	Hydrophobic	Pi–Sigma
5.31932	Hydrophobic	Alkyl
5.12768	Hydrophobic	Pi–alkyl
4.84944	Hydrophobic	Pi–alkyl
4.1316	Hydrophobic	Pi–alkyl

**Table 6 molecules-28-01833-t006:** The active residues, bond length (A0), bond types, and bond categories involved in the molecular interactions of curcumin with ACE2 receptors (7KMB, 7KNB, and 7KNH) in a complex.

Bond Length (A0)	Bond Category	Bond Type
**Curcumin–7KMB receptor in a complex**		
2.58816	Hydrogen bond	Conventional hydrogen bond
2.03326	Hydrogen bond	Conventional hydrogen bond
1.99813	Hydrogen bond	Conventional hydrogen bond
2.08162	Hydrogen bond	Conventional hydrogen bond
2.21662	Hydrogen bond	Conventional hydrogen bond
2.95087	Hydrogen bond	Conventional hydrogen bond
4.14296	Electrostatic	Pi–anion
5.1296	Hydrophobic	Pi–Pi Stacked
5.35067	Hydrophobic	Pi–alkyl
**Curcumin–7KNB receptor in a complex**		
2.07633	Hydrogen bond	Conventional hydrogen bond
2.07392	Hydrogen bond	Conventional hydrogen bond
2.71198	Hydrogen bond	Conventional hydrogen bond
2.63368	Hydrogen bond	Conventional hydrogen bond
2.60406	Hydrogen bond	Conventional hydrogen bond
3.01875	Hydrogen bond	Conventional hydrogen bond
3.71325	Hydrogen bond	Carbon hydrogen bond
4.66974	Hydrophobic	Amide–Pi Stacked
4.4217	Hydrophobic	Alkyl
3.88384	Hydrophobic	Pi–alkyl
4.12778	Hydrophobic	Pi–alkyl
**Curcumin–7KNH in a complex**		
2.68924	Hydrogen bond	Conventional hydrogen bond
2.75846	Hydrogen bond	Conventional hydrogen bond
2.39584	Hydrogen bond	Conventional hydrogen bond
2.44114	Hydrogen bond	Conventional hydrogen bond
2.66824	Hydrogen bond	Conventional hydrogen bond
3.40111	Hydrogen bond	Carbon hydrogen bond
3.56468	Hydrogen bond	Carbon hydrogen bond
3.74797	Hydrogen bond	Carbon hydrogen bond
4.00134	Electrostatic	Pi–anion
4.55541	Hydrophobic	Alkyl

**Table 7 molecules-28-01833-t007:** The composition of the prepared NLC.

F. Code	Drug (mg)	Solid Lipid %	Liquid Lipid (%)
**F1**	10	1	1
**F2**	10	2	2
**F3**	10	3	3
**F4**	10	4	4
**F5**	10	5	5
**F6**	10	1	2
**F7**	10	2	4
**F8**	10	3	6
**F9**	10	4	8
**F10**	10	5	10
**F11**	10	2	1
**F12**	10	4	2
**F13**	10	6	3
**F14**	10	8	4
**F15**	10	10	5
**F16**	10	3	1
**F17**	10	6	2
**F18**	10	9	3
**F19**	10	12	4
**F20**	10	15	5

Solid lipid and liquid lipid = 1:1, 1:2, 2:1, and 3:1%.

## Data Availability

Not applicable.

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
