# Peer review of "Development of Computational In Silico Model for Nano Lipid Carrier Formulation of Curcumin"

_molecules, 2023, doi:10.3390/molecules28041833_

Round 1
Reviewer 1 Report (New Reviewer)
In general, the article is truly interesting, with innovative information, and very useful, to be replicated in other studies, is well written, with high scientific soundness. Some comments and suggestions are given in order to improve the quality of the manuscript.
a) Molecules journal does not have a strict formatting requirement, but readers are more accustomed to finding the materials and methods section after the introduction section, please reconsider the order of your sections.
b) Please enumerated your formulas and put them in the same format, please improve the formula for the % of entrapment efficiency, please
c) for DSC protocol, add the gas used in the experiment (nitrogen?, argon?) (section 3.4)
d) Also, for section 3.4., add the software used in each characterizations techniques
e) Introduction and abstract are clear and well written, useful to resume the study.
f) Please describe in the text what represents the numbers in figures 1, 2, and 3. If you don't want to describe them, delete them from the figures
g) Please specify in the text what means the black short lines in figures 1, 2, and 3.
h) Please increase the quality of table 1, is smashed
I) Please put in the center all figures
j) Please make sure that all tables have the same format
k) For table 2, add the "units" in the particle size column
l) Please discussed more results of section 2.2. the results are good?, compared to which studies?, what is the necessary particle sizes, and which of the formulations are the best and which are the worst
m) In figure 5, you have two different FTIR spectrum formats, please improve the figure.
n) All figures have to be in the same format, please make sure all figures have the same thickness, same colors, and same representation of the samples
o) Please compare DSC results with other studies
p) Please compare section 2.4.1 with other studies, the incorporation efficiency is as expected?, better than what studies?
q) Try to connect all characterization techniques with the final application,
r) In conclusion section is missing a final reflection of your work and future perspectives, what's next?
s) I found several minimal typing errors, please review carefully your manuscript to correct them
Author Response
Dear Sir
Thank you very much for your comments to enhance the manuscript
Please find attached file
Thanks again

Reviewer 2 Report (New Reviewer)
Please see the attachment.

Author Response
Dear Sir
Thank you very much for your valuable comments, attached is the report that shows the action that was taken to each comment that is highlighted in yellow, however, the manuscript have blue highlights as well that represent the response to the first reviewer
Thanks again

Reviewer 3 Report (New Reviewer)
Organisation of the article should be changed. Results and discussion should follow materials and methods.
The predicted and actual solubility and stability should be presented in a comparison table.
Include a paragraph correlating the EE%, particle size, PDI, and zeta potential.
Cite few recent references with similar studies.
Author Response
Dear Sir
Attached is the report that contain all the revisions that were made to the manuscript
Thank you for your comments that enhance the manuscript

Round 2
Reviewer 1 Report (New Reviewer)
In my point of view, author successfully attend all my comments and suggestions
This manuscript is a resubmission of an earlier submission. The following is a list of the peer review reports and author responses from that submission.
Round 1
Reviewer 1 Report
The paper report the study of curcumin based formulation of NLC using an in silico method. Once selected the best components and preparation method, pure components and NLC were characterized at the solid state with DSC, XRD, FTIR while NLC were analyzed in terms of drug content, particle size, zeta potential, PDI, AFM, drug release, in vitro and ex-vivo permeation studies and molecular docking studies of Anti-Viral activity of curcumin NLC against SARS-CoV-2.
The work has many interesting results, unfortunately the quality of English is very low and there are many typos that make reading very difficult.
Some suggestions to improve the quality of the work:
Which kind of experimental design was used?
FTIR, DSC and XRD analysis: loaded NLC should be compared with empty NLC or pure components.
DSC : in the thermographs endo direction should be reported, in figure 10 what is the peak at 240 ° C attributed to?
Too many figures are reported XRD, DSC and FTIR figures can be summarized, while figures 13 and 14 can be deleted
Author Response
Dear Dr
Thank you very much for your comments
Please find the attached file
Regards

Reviewer 2 Report
The authors presented formulation and investigation of a novel curcumin nano lipid, where a computational molecular docking method was applied to find optimum solid and liquid lipids for NLCs.
Overall the methods used for the study were technically sound, they were not well structured, clear, the authors’ claims were not fully supported by the data presented. Some sentences are confused and incomprehensible.
Some of my specific questions and comments are the follows:
1. The abstract is too long (over the 200 words limit).
2. In the introduction, Authors mentioned the degradability of curcumin without the reason of its instability. For example, can high temperatures be used during the formulation of NLCs?
3. Description of some methods is completely missing (DSC, XRD, HPLC, spectrophotometry).
4. For the particle size, the method description cannot be interpreted: "Non-invasive backscatter technology was used to detect light scattering at 173°C, measuring approximately 0.6 nm to 6 m." Also, why did you filter the samples before particle size analysis?
5. The spectrophotometric technique used cannot be considered selective, but since there is no description for it, it cannot be evaluated.
6. What is the difference between "in vitro permeation study" and "ex vivo everted gut sac method" in the presented work, it is not clear why both methods were used.
7. Sometimes he talks about a curcumin solution, sometimes a suspension, but these are constantly mixed, it doesn't matter at all what form the active ingredient is in.
8. The figures are completely different from a formal point of view (even within one method, see eg. FTIR).
9. From the DSC tests, the presence of the active substance as a molecularly dispersed form cannot be confirmed, as described, since it can also dissolve during the measurement (as the result of heating).
10. There are several incomprehensible parts in the manuscript, just some examples:
„It shows (figure 13) the particle size of the F1 formulation (99.648.64), whereas shows the PDI of 0.1920.01, which is near to zero.”
„Scheme 1. to be - 42.30.±01 mV, figure 14.”
11. Figure 15 is missing.
Author Response
Der Dr.
Thank you very much for your comments to enhance the manuscript
Please find the attached file
Regards

Round 2
Reviewer 1 Report
Taking into account the revision, the paper can be accepted in the present form
Reviewer 2 Report
I thank the authors for their answers, but I think they are still not satisfactory, the quality of the article is still not up to the standard of the journal.
My first comment was on the length of the abstract, as the maximum length of the abstract in this journal is 200 words, so I indicated that the current version of more than 500 words was not adequate. The authors did not make any changes, but put a question mark as response.
Some of the method descriptions have been replaced, but their depth is not sufficient (e.g. FT-IR method: scan number, sample preparation, module used) and the HPLC method is still missing.
In the method descriptions I indicated that there were unintelligible sentences and data, which have not been corrected in the revised version ("detect light scattering at 173°C" the temperature was 173°C??? or it is just 173°, or "measuring approximately 0.6 nm to 6 m" 6 m as particle size???).
For the thermal sensitivity of the active substance, the authors stated in their reply that the active substance is exposed to high temperatures for a short time. I do not believe this to be the case, the drug is dissolved in the lipid melt, then stirred at high temperature for 15 minutes, and HHPH was applied for a further period of time at high temperature. This is not considered a short time in my opinion.
The quality of the figures has still not improved, the XRD plots usually show 2 theta on the x axis, in this case perhaps only the theta is readable (quite poorly visible).
The authors have not improved the English of the manuscript, there are still many errors and difficult to understand parts.